# Effects of Ultrafine Single-Nanometer Oxygen Bubbles on Radiation Sensitivity in a Tumor-Bearing Mouse Model

**DOI:** 10.3390/ijms23126838

**Published:** 2022-06-20

**Authors:** Navchaa Gombodorj, Takehiko Yokobori, Nobutoshi Mutsuki, Bilguun Erkhem-Ochir, Haruka Okami, Takayuki Asao, Hiroshi Saeki, Ken Shirabe, Dai Yamanouchi

**Affiliations:** 1Division of Integrated Oncology Research, Gunma University Initiative for Advanced Research (GIAR), 3-39-22 Showamachi, Maebashi 371-8511, Japan; navch0025@gmail.com (N.G.); bilguunerkhemochir@gmail.com (B.E.-O.); 2Department of Radiation Oncology, National Cancer Center, Ulaanbaatar 13370, Mongolia; 3Division of Referral and Primary Healthcare Services, Ministry of Health, Ulaanbaatar 14210, Mongolia; 4Sigma Technology Inc., 1-11-29 Sotono, Hitachinaka 312-0053, Japan; mec104@mutsuki.co.jp; 5Department of General Surgical Science, Graduate School of Medicine, Gunma University, Maebashi 371-8510, Japan; m2220010@gunmau.ac.jp (H.O.); h-saeki@gunma-u.ac.jp (H.S.); kshirabe@gunma-u.ac.jp (K.S.); 6Gunma University Center for Mathematics and Data Science, 4-2 Aramaki, Maebashi 371-8510, Japan; asao@gunma-u.ac.jp; 7Division of Vascular Surgery, Department of Surgery, University of Wisconsin School of Medicine and Public Health, Madison, WI 53726, USA

**Keywords:** ultrafine nanobubble oxygen, hypoxia, HIFs, radiation sensitizer, tumor-bearing mouse model

## Abstract

Radiation therapy against cancer cells often causes radiation resistance via accumulation of hypoxia-inducible factor 1 subunit alpha (HIF-1α) under hypoxic conditions and severe side effects. Radiation sensitizers without side effects are required to overcome hypoxia-induced radiation resistance and decrease radiation-related side effects in patients with refractory cancer. We previously developed oxygen nanobubble water (NBO_2_ water) and demonstrated that it suppresses hypoxia-induced radiation resistance in cancer cell lines within the single-nanometer range. This study aimed to elucidate whether NBO_2_ water could act as a radiosensitizer via regulation of HIF-1α in a tumor-bearing mouse model. Six-week-old female BALB/c mice subcutaneously injected with tumor cells received control water or NBO_2_ water for 28 days, after which biochemical examinations and radiation treatment were performed. Hypoxic tumor regions were detected immunohistochemically. We found that NBO_2_ water sensitized radiation reactivity in the xenografted tumors. Notably, NBO_2_ water administration downregulated the accumulation of HIF-1α in xenografted tumors and did not affect the vital organs of healthy mice. The combination of radiation and single-nanometer NBO_2_ water without severe side effects may be a promising therapeutic option to improve radiation sensitivity in cancer patients without tolerance to invasive treatments.

## 1. Introduction

Radiation therapy is often used for destroying cancer cells and as palliative therapy [1,2], but often induces radiation resistance; indeed, high-dose radiation may cause acute and chronic severe side effects, including lethal radiation-induced pneumonitis and fibrosis [3,4]. Therefore, the development of radiation sensitizers without side effects is urgently needed to overcome radiation resistance and decrease radiation-related lethal severe side effects in patients with refractory cancer.

Growing tumors result in complex tumor microenvironments with low-oxygen areas because of the severe structural abnormalities of tumor microvessels [5,6]. Hypoxia restricts the physiological functions of organs, tissues, and cells, including cancer cells. To survive in hostile hypoxic environments, hypoxia-inducible factors (HIFs) can function as essential regulators of cellular oxygen homeostasis and hypoxia adaptation in tumor tissues [7,8,9]. HIFs are heterodimers, including one of three HIF-α subunits, HIF-1α, HIF-2α, or HIF-3α (and one constitutive HIF-1β subunit known as ARNT), which form HIF-1, HIF-2, and HIF-3 transcriptional complexes, respectively [10]. Among their subunits, HIF-1α has been well studied and is known to activate several downstream genes transcriptionally related to angiogenesis, anaerobic metabolism, and resistance to radiation in cancer tissues [11]. Many researchers have reported the importance of an accumulation of HIF-1α in clinical cancer tissues, causing radiation resistance, cancer aggressiveness, and poor prognosis [10]. However, an HIF-1α-targeting strategy may induce deleterious side effects in non-cancerous tissues [12,13,14]. Therefore, the development of therapeutic tools targeting HIF-1α without side effects is urgently needed for refractory cancers with radiation resistance.

Previously, we reported the development of oxygen nanobubble (NBO_2_) water in the single-nanometer range, which was produced by Sigma Technology Inc. (Hitachinaka, Japan) [15]. Our in vitro analysis clearly showed that NBO_2_ water suppressed hypoxia-induced radiation resistance in cancer cell lines via the downregulation of HIF-1α accumulation under hypoxic conditions. Notably, NBO_2_ water did not affect the in vitro cell viability and radiation reactivity of either cancer or non-cancerous cell lines under normoxic conditions, suggesting that NBO_2_ water might have the potential to overcome hypoxia-induced radiation resistance without side effects, even in an animal model. Therefore, we designed this study to examine the effects of NBO_2_ water in a tumor-bearing mouse model.

The purpose of this study was to elucidate whether NBO_2_ water could act as a radiosensitizer via regulation of HIF-1α in a tumor-bearing mouse model and whether it has any side effects consistent with previous in vitro experiments.

## 2. Results

### 2.1. Suppression of Hypoxic Conditions in Cancer Cell Lines Treated by the NBO_2_ Media

A hypoxic MAR probe was used to analyze the effect of NBO_2_ water against living EBC-1 cells under both normoxic and hypoxic conditions. The fluorescence intensity of water media-treated EBC-1 cells under hypoxic conditions was higher than that under normoxic conditions (Figure 1, left panel). Under normoxic conditions, the fluorescence intensity of NBO_2_ media-treated cells was not altered compared to that of water media. However, the hypoxia-induced fluorescent intensity of EBC-1 cells was downregulated by NBO_2_ media-treated EBC-1 cells compared to that in water media (Figure 1, right panel).

Weak fluorescence intensity was detected by the hypoxic MAR probe in lung cancer cell line EBC-1 treated with water and NBO_2_ media under normoxic conditions. Lower panel: Fluorescence intensity by MAR was strongly detected in cells cultured in water media under hypoxic conditions; however, fluorescence induction by the hypoxic condition was weaker in the NBO_2_ media than in the water media.

### 2.2. Safety Verification of NBO_2_ Water in Mice

To analyze the safety of NBO_2_ water, healthy BALB/c mice received either control water or NBO_2_ water in a free-drinking environment for 28 days (Figure 2A). After 28 days, the administration of NBO_2_ water did not influence mouse appearance, body weight, red blood cell count, or serum biochemical examinations (Figure 2B,C).

Next, we compared the appearance and histology of vital organs in mice to assess whether the administration of NBO_2_ water causes morphological and pathological abnormalities in vital organs. As a result, the macroscopic size and histological findings of vital organs in the mice showed no apparent difference between the control water and NBO_2_ water groups (Figure 3A,B).

### 2.3. Effect of NBO_2_ Water with Radiation in a Xenograft Mouse Model

A subcutaneous xenograft mouse model was established to determine the combined effects of radiation and free drinking of NBO_2_ water for 28 days (Figure 4A,C). Consistent with our previous in vitro data [15], we confirmed that the combination of NBO_2_ water and radiation was the most effective treatment compared to the control water groups in both tumor xenograft types, EBC-1 and HCT116 (Figure 4B,D). Interestingly, the administration of NBO_2_ water even without radiation showed a tendency for tumor growth suppression compared to the water-only group.

### 2.4. Suppression of Hypoxic Condition and HIF-1α Expression by NBO_2_ Administration in Xenografted Tumors

The expression of HIF-1α in the xenografted tumors was downregulated in the NBO_2_ water groups compared to that in the water groups (Figure 5A). The immunohistochemical analysis demonstrated the same tendency of HIF-1α alteration between the NBO_2_ and water groups as in the Western blotting results (Figure 5B). To validate the effect of NBO_2_ water against hypoxic conditions in xenografted tumors, we stained for the hypoxic marker pimonidazole. Area fractions showing pimonidazole staining were identified as hypoxic areas. Since the intensity of pimonidazole staining was weaker in the NBO_2_ water group than in the control group, the NBO_2_ water group was considered less hypoxic (Figure 5C).

To validate the hypoxia-related signal alteration in each group, we evaluated the immunohistochemical staining of carbonic anhydrase IX (CA IX) as a representative hypoxia-related downstream pathway regarding radiation resistance in cancer cells [16]. As a result, the expression levels of CA IX in the NBO_2_ water and NBO_2_ + IR groups were lower than those in the other groups (Figure 6).

## 3. Discussion

Radiation treatments can cause acute tissue effects (mucositis, radiodermatitis, pneumonia, and others) based on DNA damage, inflammation, and vascular depletion, and late adverse tissue effects (hardening of irradiated organs such as lung or breast tissues; malabsorption, small bowel narrowing, radiation-induced secondary carcinogenesis, and others) due to fibrosis and vascular changes associated with ischemia and hypoxia. Among these various changes in the radiation treatment process, hypoxia/ HIF-1α levels have been involved in fibrosis and vascular changes [4,17,18]. HIF-1α activates TGF-β signaling and integrin expression in stromal cells, causing fibrosis in vital organs. Furthermore, hypoxia/HIF-1α induces the secretion of CXCL12, VEGF, and ANG1/2, leading to vascular changes, such as fragile tumor angiogenesis, through the recruitment of vascular endothelial cells. Therefore, the therapeutic strategy of suppressing hypoxia/HIF-1α in tumor tissues using NBO_2_ water may affect both radiation sensitivity and radiation-related side effects.

In this study, we found that NBO_2_ water sensitized radiation reactivity via suppression of radioresistance-related HIF-1α and CA IX expression in a tumor-bearing mouse model, consistent with our previous in vitro data [15]. Notably, NBO_2_ water administration for 28 days downregulated the accumulation of HIF-1α in the xenografted tumors and did not affect the vital organs or biochemical examinations in healthy wild-type mice. We did not observe obvious side effects (appearance, body weight alteration, or treatment-related deaths as acute side effects) between radiation with or without NBO_2_ water treatment. This may be because our radiation settings had already been adjusted to complete the animal experiments. As our study was conducted to evaluate the combined effects of NBO_2_ water and radiation in a tumor-bearing mouse model, we could not obtain data on the relationship between high-dose radiation-related late complications and NBO_2_ administration under our experimental conditions. Future studies are needed to elucidate the influence of NBO_2_ administration on radiation-related side effects. On the other hand, amifostine has been approved as a clinical radiation protector to reduce the radiation-related side effect in normal tissues [19]. Therefore, we expect that amifostine combined with NBO_2_ water could allow for higher radiation doses, thereby enhancing the therapeutic effect without increasing radiation-related side effects.

Surgical operation, radiation, and anticancer drugs are recognized as essential treatment methods in current cancer care [20,21,22]. However, some patients with poor general condition cannot undergo highly invasive treatments. In clinical practice, despite the existence of promising therapeutic options such as surgery or chemotherapy, patients with advanced or recurrent cancer and patients without tolerance to invasive treatments are often not provided with any other alternatives and rather offered the best supportive care [23,24]. This study focused on NBO_2_ water, which suppressed intratumor hypoxia/HIF-1α and showed radio-sensitizing effects without side effects. Of particular interest to us was that the NBO_2_ water group, even the group with NBO_2_ water alone, showed a certain degree of antitumor effects compared to the control water group. As mentioned above, NBO_2_ water targets intratumoral hypoxia/HIF-1α, which is known to cause therapeutic resistance to radiation and various anticancer drugs [25,26,27]. Thus, overcoming hypoxia/HIF-1α is a promising therapeutic strategy for hypoxia-related refractory cancer patients. If targeting intratumoral hypoxia/HIF-1α using NBO_2_ water without side effects is made possible, treatment of high-risk cancer patients who cannot tolerate surgery or systemic chemotherapy would be feasible. We hope that treatment with NBO_2_ alone or in combination with NBO_2_ water with dose-reduced chemotherapy or radiation will improve the quality of life and prolong the prognosis in cancer patients who cannot tolerate highly invasive treatments.

Free drinking water was selected as a general and straightforward administration route of NBO_2_ water in the animal model. However, oral intake is often impossible for patients with advanced cancer with poor conditions. Therefore, in the future, we would like to investigate the relationship between the hypoxia/HIF-1α suppression effect and the route of administration of NBO_2_ water, such as intravenous or intraperitoneal injections.

## 4. Materials and Methods

### 4.1. Production of NBO_2_ Water

NBO_2_ water with ultrafine oxygen bubbles of single-nanometer size was provided by Sigma Technology Inc. and produced by their original device based on patented technology [15]. Briefly, oxygen and pure water were mixed at 0.4 MPa and pushed out of a nozzle. Oxygen-containing water collided at a high speed to produce the NBO_2_ water with single nanometer-sized oxygen bubbles.

### 4.2. Cell Lines

Human lung cancer cell line EBC-1 and human colon cancer cell line HCT116 were purchased from the RIKEN Cell Resource Center of Biomedical Research (Tsukuba, Japan). Culture media were prepared using RPMI1640 powder (Wako, Osaka, Japan) and dissolved in control water (water) or water with ultrafine oxygen bubbles (NBO_2_). Cells were cultured in filtered (0.22 μm) RPMI1640 supplemented with 10% fetal bovine serum and 1% penicillin and streptomycin antibiotics and incubated at 37 °C and 5% CO_2._ For hypoxia induction, the cells were incubated under hypoxic conditions (1% O_2_) using a BIONIX-1 hypoxic culture kit (Sugiyamagen, Tokyo, Japan).

### 4.3. Hypoxia Detection by Fluorescent Hypoxic Probe

Cells were seeded at a density of 4 × 10^3^/100 μL per well in 6-well plates. After 48 h, the culture medium was changed to media with or without NBO_2_ and the plates were incubated with 1 μM hypoxic probe MAR (Goryo Chemical, Inc., Sapporo, Japan) under normoxic or hypoxic conditions. After 24 h of incubation, the fluorescence signal was detected using an All-in-One BZ-X710 fluorescence microscope (KEYENCE Corporation).

### 4.4. Protein Extraction and Western Blotting

Protein extraction was performed using lysis buffer (10% glycerol, 10 mM Tris-HCl (pH7.5), 1 mM EDTA, 400 mM NaCl, 0.5% NP40, 4 μg/mL aprotinin, PMSF, proteasome inhibitor MG-132, and 1mM DTT). Total protein (10 µg) was electrophoresed on a 10% polyacrylamide gel and then electroblotted at 300 mA for 90 min on a nitrocellulose membrane (Invitrogen, Waltham, MA, USA). Western blotting was used to confirm the expression of HIF-1α and β-actin proteins, which were detected using anti-HIF-1α rabbit polyclonal antibody (1:1000) (Cell Signaling Technology, #3716, Danvers, MA, USA) and β-actin mouse monoclonal antibodies (1:1000) (A5316; 1:1000; Sigma, St. Louis, MO, USA), respectively. β-actin was used as a loading control. The signals were detected using the ECL Select Western Blotting Detection System (GE Healthcare Life Sciences, Chicago, IL, USA) and Image Quant LAS 4000 (GE Healthcare Life Sciences).

### 4.5. Safety Assessment of the NBO_2_ Water in Mice

Six-week-old female BALB/c mice were used in this experiment. Control water or NBO_2_ water was administered via free oral drinking for 28 days. The evaluation of the general condition and body weight in each group was continued until day 28 and the vital organs were harvested for macroscopic and histological evaluation on day 28. Biochemical examinations were performed by Oriental Yeast Co., Ltd., Tokyo, Japan.

### 4.6. Nude Mouse Xenograft Model

Six-week-old female BALB/c nu/nu nude mice were subcutaneously injected with 5 × 10^6^ EBC-1 and 5 × 10^6^ HCT116 cells. Tumor volume was calculated using the following formula: volume = S^2^ × L/2, where S is the shortest length of the tumor (mm) and L is the longest length of the tumor (mm). Tumor volumes and body weights were determined every 4 days. After the tumor volume reached 100 mm^3^, the mice were randomly divided into four groups: water group as control, NBO_2_ water alone, radiation alone, and radiation plus NBO_2_ water. Each group consisted of six mice. Water or NBO_2_ was administered via free oral drinking for 28 days. Radiation of 3 Gy (HCT116) or 4 Gy (EBC-1) was administered on days 4 and 8. Tumor volume and mouse body weight were evaluated until day 28, and xenograft tumors were harvested for further analysis on day 28. The animal experiments were approved by the Review Committee on Animal Use of Gunma University (approval number 15-046, 7 September 2015).

### 4.7. Radiation Treatment against Xenograft Tumors

We performed local radiation treatment of xenograft tumors with 6 mm lead shielding using a subcutaneous xenograft mouse model. Radiation was performed at a dose of 3 or 4 Gy for HCT116 and EBC-1 cell lines, respectively (on days 4 and 8) using a TITAN-225S (Shimadzu Mectem, Inc., Tokyo, Japan) at 200 kV and 14.6 mA with Al 0.5 mm and Cu 0.5 mm filtration at a distance of 47.3 cm from the target.

### 4.8. Detection of Hypoxic Conditions in Xenograft Tumors

Pimonidazole was used as a marker of tumor hypoxia. Following the manufacturer’s instructions, pimonidazole (Hypoxyprobe-1 Omni Kit, #HP3-100) at a dose of 60 mg/kg body weight was administered to the tumor-bearing mice intraperitoneally 30 min before tumor excision. Hypoxic tumor regions were detected immunohistochemically. Hematoxylin was used as a counterstain. Area fractions showing pimonidazole staining were identified as hypoxic areas.

### 4.9. Immunohistochemistry

Four-micron sections were cut from paraffin blocks of resected xenograft tumors and each section was mounted on a silane-coated glass slide, deparaffinized, and soaked in 0.3% H_2_O_2_/methanol for 30 min at room temperature to block endogenous peroxidases. The sections were then heated in boiling water using an Immunosaver (Nishin EM, Tokyo, Japan) for 45 min at 98 °C. Non-specific binding sites were blocked by incubating the sections with Protein Block Serum-Free (DAKO, Santa Clara, CA, USA) for 30 min. Anti- HIF-1α antibody (Novusbio, #NB100-479, Littleton, Co, USA), anti-pimonidazole antibody (Hypoxyprob-1 Omni Kit, #HP3-100), and CA IX antibody (Abcam, ab15086, Cambridge, UK) were each applied in dilutions of 1:100 (HIF-1α and pimonidazole) and 1:500 (CA IX) for 24 h at 4 °C. The primary antibody was visualized using the Histofine Simple Stain MAX-PO (Multi) Kit (Nichirei, Tokyo, Japan) according to the manufacturer’s instructions. The chromogen 3,30-diaminobenzidine tetrahydrochloride was used as a 0.02% solution containing 0.005% H_2_O_2_ in 50 mM ammonium acetate–citrate acid buffer. The sections were lightly counterstained with Mayer’s hematoxylin and mounted. Negative control specimens were incubated without primary antibodies and no detectable staining was evident.

### 4.10. Statistical Analysis

Data are expressed as the mean ± standard deviation for continuous variables. The differences between groups were compared using JMP Pro 15 software (SAS Institute, Cary, NC, USA). The Mann-Whitney U-test and Steel–Dwass test were used to compare the groups. A probability (*p*) value of less than 0.05 was considered statistically significant.

## 5. Conclusions

We previously reported on the development of NBO_2_ water with single-nanometer-range oxygen that could improve radiation sensitivity via the suppression of HIF-1α in an in vitro analysis of cancer cell lines. This study demonstrated the safety, antitumor effect, potential as a radiation sensitizer, and tumoral hypoxia/HIF-1α suppression effect of oral administration of NBO_2_ water in vivo in a tumor-bearing mouse model.

## 6. Patents

Tachibana Y., Tachibana K., Harada K., Sasajima S., Honma K., Matsumoto Y. Micro-/nanobubble-generating method, -generating nozzle, and -generating device. Sigma Technology, Inc. JP, issued 23 July 2014. JP5555892B2.

## Figures and Tables

**Figure 1 ijms-23-06838-f001:**
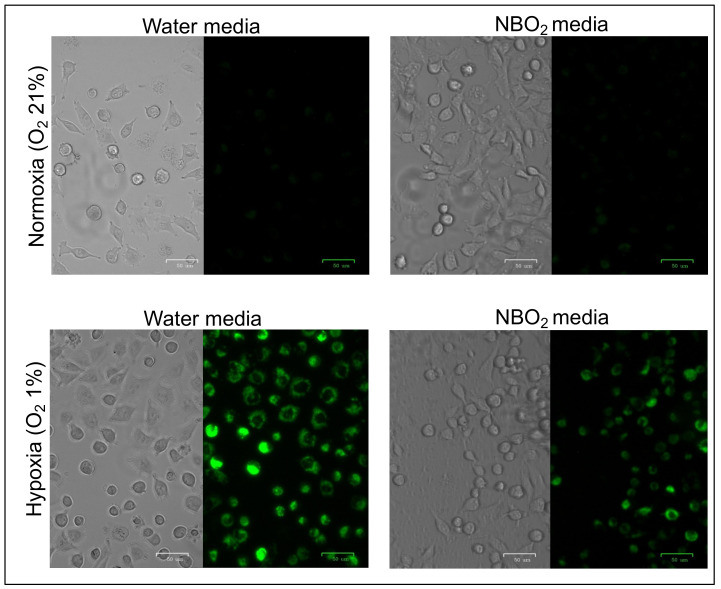
Suppression effect of hypoxia/HIF after treatment with NBO_2_ water.

**Figure 2 ijms-23-06838-f002:**
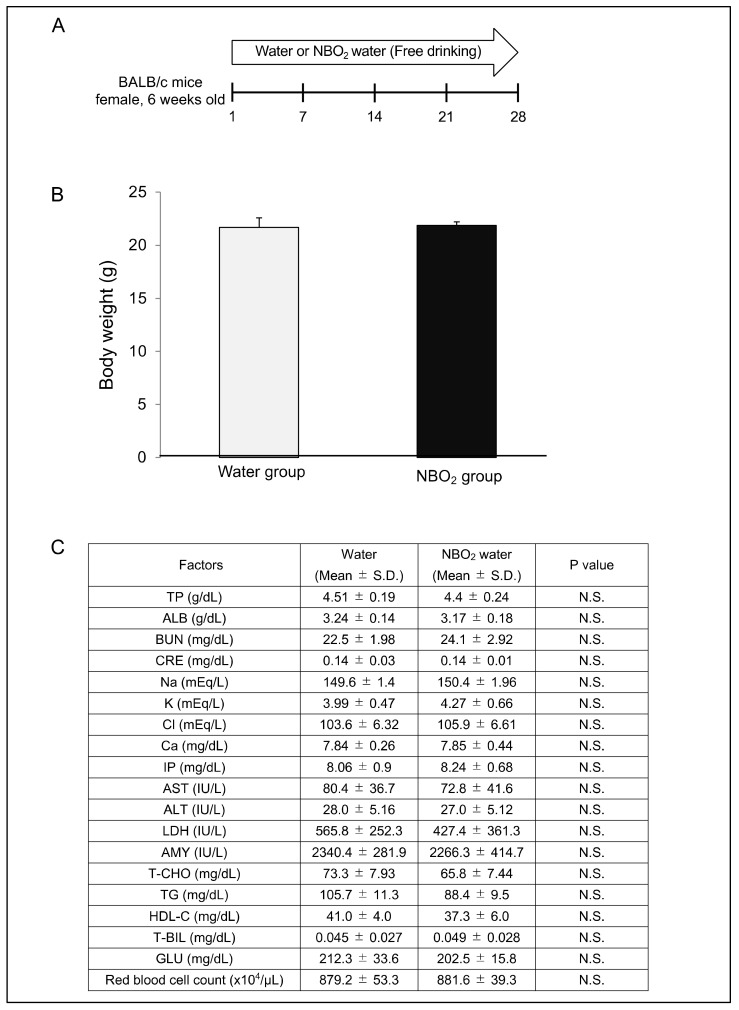
Safety of NBO_2_ water in mice. (**A**) Schedule of this study. Oral intake of water and NBO_2_ water in BALB/c mice was carried out from day 1 to day 28. Blood samples were harvested at day 28. (**B**) The body weight of the mice in water (*n* = 7) and NBO_2_ water (*n* = 10) groups at day 28. (**C**) Serum biochemical examinations and red-blood-cell count in blood from BALB/c mice treated with water (*n* = 7) and NBO_2_ water (*n* = 10) over 28 days.

**Figure 3 ijms-23-06838-f003:**
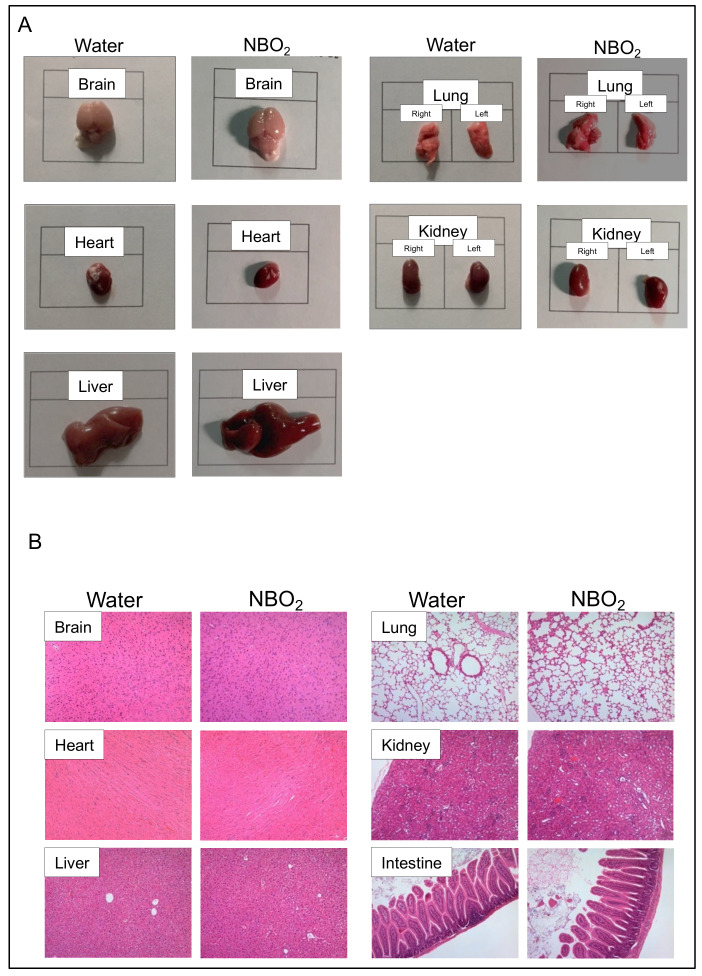
Macroscopic and histological findings of organs in healthy mice receiving free drinking water or NBO_2_ water for 28 days. (**A**) Macroscopic view of organs in mice treated with water and NBO_2_ water. (**B**) Histological findings of organs in mice treated with water and NBO_2_ water (original magnification × 100).

**Figure 4 ijms-23-06838-f004:**
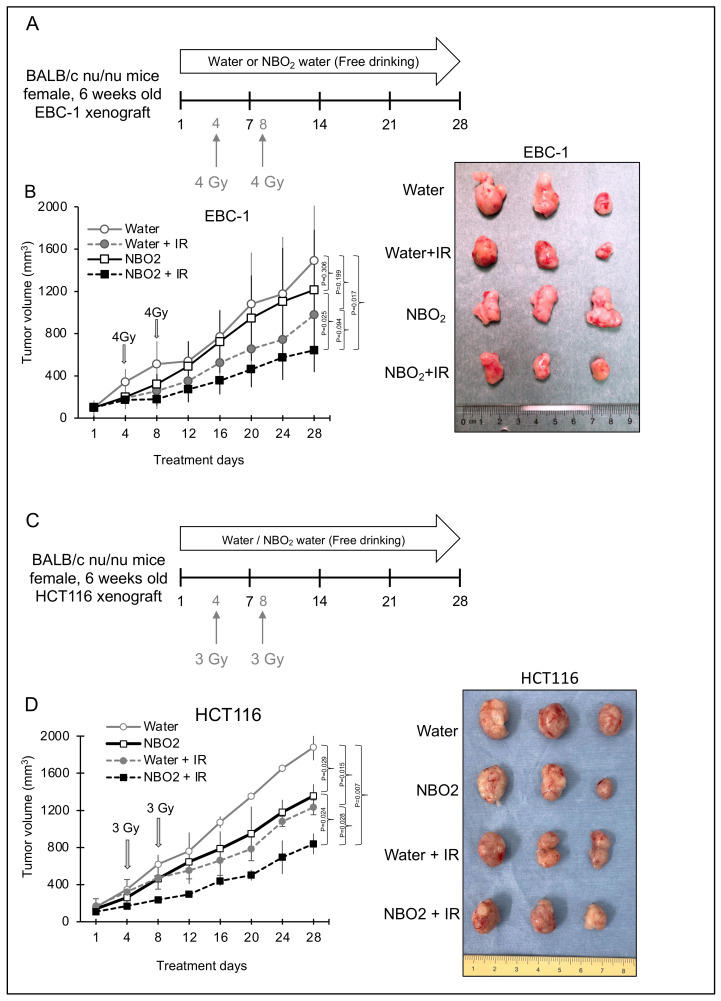
Therapeutic effect of drinking NBO_2_ water and radiation in a tumor-bearing mouse model using EBC-1 and HCT116 cells. (**A**,**C**) Schedule of these studies. Oral intake of water and NBO_2_ water was carried out from day 1 to day 28. Radiation (EBC-1, 4 Gy/fraction: HCT116, 3 Gy/fraction) against xenografted EBC-1 tumors on days 4 and 8. The xenograft tumors were harvested on day 28. (**B**) Left panel: tumor volume in an EBC-1 xenograft model in water, water + radiation (IR), NBO_2_, and NBO_2_ + IR groups. Right panel: representative images of harvested tumors in each group taken on day 28. (**D**) Left panel: tumor volume in an HCT116 xenograft model in water, water + radiation (IR), NBO_2_, and NBO_2_ + IR groups. Right panel: representative images of harvested tumors in each group taken on day 28.

**Figure 5 ijms-23-06838-f005:**
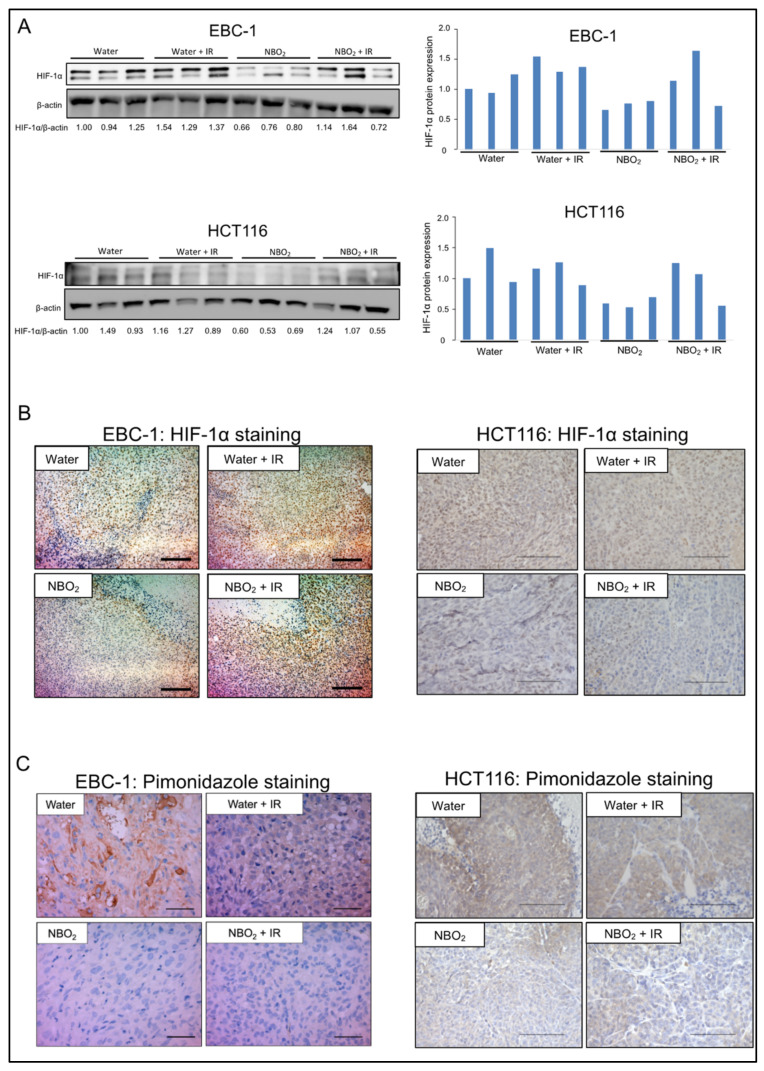
Suppression effect of HIF-1α and hypoxic condition by NBO_2_ water in EBC-1 and HCT116 xenografted tumors. (**A**) Protein expression of HIF-1α in EBC-1 tumor xenografts of each group detected by Western blotting. These proteins were normalized to the level of β-actin expression. Intensity was measured using Image J software. (**B**) Representative immunohistochemical staining of HIF-1α in EBC-1 and HCT116 xenografted tumors of each group (EBC-1, scale bar 200 μm; HCT116, scale bar 100 μm). (**C**) Histological section of pimonidazole staining in EBC-1 and HCT116 xenografted tumors of each group in 28 days (EBC-1, scale bar 50 μm; HCT116, scale bar 100 μm).

**Figure 6 ijms-23-06838-f006:**
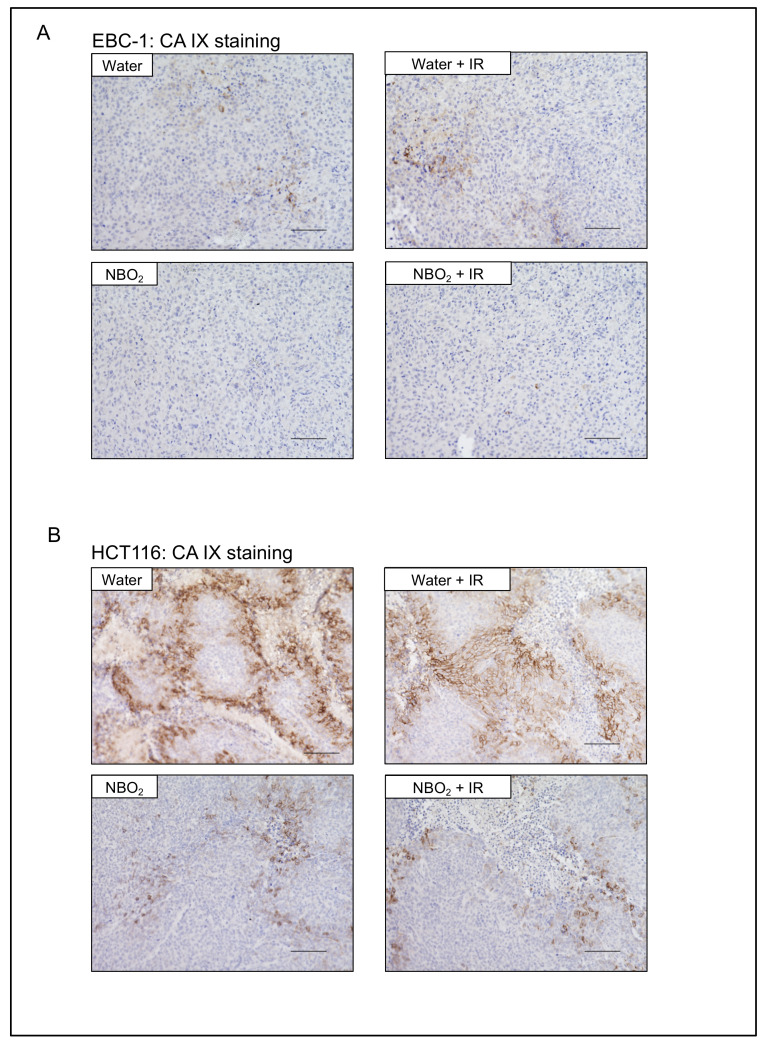
Hypoxia-related carbonic anhydrase IX (CA IX) expression in EBC-1 and HCT116 xenografted tumors of each treatment group. (**A**,**B**) Representative immunohistochemical staining of CA IX in EBC-1 and HCT116 xenografted tumors of each group (scale bar 100 μm).

## Data Availability

The data presented in this study are available on request from the corresponding author.

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
