# Peer review of "Effects of Ultrafine Single-Nanometer Oxygen Bubbles on Radiation Sensitivity in a Tumor-Bearing Mouse Model"

_ijms, 2022, doi:10.3390/ijms23126838_

Round 1

Reviewer 1 Report

Title of paper: Effects of ultrafine single-nanometer oxygen bubbles on radiation sensitivity in a tumor-bearing mouse model

The title and abstract cover the main aspects of the research. The introduction provides short background and summarized information about the study. The introduction needs some minor modifications. Previous characterization studies should be presented in this section. The methods are clear and replicable and the presented results are according to the applied methods. Also, the findings described by the author correlate with the presented results. However, the claimed conclusion is not supported by the findings. The methods need to be very well explained starting from the conceptualization and design. Here are some aspects to be considered:

1.       Please, explain the reason to use EBC-1 cell line Human lung cancer cell line EBC-1, and human colon cancer cell line HCT116. What is the feature of these cell lines than make them suitable for this research? Present your ethical approval on animal management.

2.       Why do the authors use water? Is the isotonicity affected? Cells in figure 1 seem affected by this condition.

3.       Was the colloidal stability demonstrated?, How it was measured?. Cells in figure 1 seem affected by this condition.

4.       How do measure the dose of the treatment? Why the size of tumors were directly correlated with the treatment if the experiment was not designed to take into account critical variables?

5.       Supplementary figures should be attached as a different document

6.       Please, review the following sentence in the abstract: “We found that NBO2 water sensitized radiation reactivity in the mice”.

7.       Explain the biodistribution of NBO2 water, how did reach the tumor? What is the specific uptake mechanism within the tumor?

8.       Please, explain the mechanism by which, the hypoxia-induced-fluorescent intensity of EBC-1 cells was downregulated by NBO2 media-treated EBC-1 cells compared to that in water media. Why is this specific?

9.       Please, show the biodistribution profile, and the histopathological analysis in the rest of the organs, kidney, spleen, lungs, and liver?.

10.   How was the elimination pathway?

11.   Explain the criteria for choosing dose radiation of 3 Gy (HCT116) or 4 Gy (EBC-1) on days 4 and 8. It seems that was a part of a fractionated treatment.

Author Response

Reply to Reviewer #1

Query 1:

Please, explain the reason to use EBC-1 cell line Human lung cancer cell line EBC-1, and human colon cancer cell line HCT116. What is the feature of these cell lines than make them suitable for this research? Present your ethical approval on animal management.

Answer 1:

Thank you for these suggestions. We selected the two cell lines derived from lung squamous cell carcinoma and colon cancer because these diseases have been treated by radiation as a neoadjuvant treatment in the clinic. Moreover, we have some experience in performing the xenograft tumor models treated by radiation using these cell lines in another research project. Therefore, we selected these cell lines in this study.

According to this comment, we added the approval information in the Materials and Methods and in the Institutional Review Board Statement as follows:

"The animal experiments were approved by the Review of Committee on Animal Use of Gunma University (approval number: 15-046). "

Query 2:

Why do the authors use water? Is the isotonicity affected? Cells in figure 1 seem affected by this condition.

Answer 2:

I apologize for the confusing description caused by the lack of explanation in Figure 1. We made culture media from RPMI1640 media power and used water with or without NBO2. We did not use mere water for this experiment.

To avoid misunderstandings, we changed “water and NBO2” to “water media and NBO2 media” in Figure 1.

Query 3:

Was the colloidal stability demonstrated?, How it was measured?. Cells in figure 1 seem affected by this condition.

Answer 3:

In previous our report, we used the cryoelectronic microscopy to visualize and measure the single nanometer bubbles because we could not see them by the normal microscopic analysis. Moreover, we have confirmed that the oxygen nanobubbles are stable for several months by using the light scattering method.

Query 4:

How do measure the dose of the treatment? Why the size of tumors were directly correlated with the treatment if the experiment was not designed to take into account critical variables?

Answer 4:

Thank you for this pertinent comment. The present study is a pilot study that demonstrates the radiosensitizer effect and safety of our oxygen nanobubble water on living animals. Therefore, we wanted to administer as much oxygen nanobubble water as possible to experimental tumor-bearing mice models.

We cannot administer excessive amounts of the solution to the mice for animal welfare reasons. If intraperitoneal injection were selected, the standard dosage would be about 20 mL/kg, and repeated daily administration would be difficult due to procedure-related complications. On the other hand, the intravenous injection can be administered at about 5 mL/kg, which is even lower than the intraperitoneal injection, and daily administration is not easy to perform.

We believe that the oral drinking chosen for this study allowed mice to safely ingest 4 to 7 mL/day of nanobubbles water daily. Of course, we agree with the referee's advice. In the future, we should show the relationship between the exact NBO2 treatment dosage and efficacy in the animal model for potential clinical trials.

Query 5:

Supplementary figures should be attached as a different document

Answer 5:

According to this comment, we have moved the supplementary figures to different document.

Query 6:

Please, review the following sentence in the abstract: “We found that NBO2 water sensitized radiation reactivity in the mice”.

Answer 6:

Thank you for this comment. We apologize sorry that our previous description was incorrect. We revised the sentence as follows:

"We found that NBO2 water sensitized radiation reactivity in the xenografted tumors."

Query 7:

Explain the biodistribution of NBO2 water, how did reach the tumor? What is the specific uptake mechanism within the tumor?

Answer 7:

Thank you for this important comment. We think that the biodistribution of the NBO2 water is a crucial factor in clarifying the action mechanism of the NBO2 water against xenograft tumors. We have already attempted an electronic microscopy experiment to visualize the intracellular NBO2 particles; however, we could not detect them. We hope to clarify this critical information in future studies.

Query 8:

Please, explain the mechanism by which, the hypoxia-induced-fluorescent intensity of EBC-1 cells was downregulated by NBO2 media-treated EBC-1 cells compared to that in water media. Why is this specific?

Answer 8:

In this experiment, we used MAR fluorescent probe for live-cell imaging. It is known that the MAR probe under hypoxic conditions is reduced by intracellular reductase and emits fluorescence. Therefore, we believe that our NBO2 media inhibits fluorescence intensity by modulating intracellular hypoxia and decreasing reductase activity.

Query 9:

Please, show the biodistribution profile, and the histopathological analysis in the rest of the organs, kidney, spleen, lungs, and liver?

Answer 9:

As mentioned above, we apologize for the inability to obtain direct evidence showing the biodistribution of intracellular NBO2particles. Further studies are needed to establish methods for specifically visualizing the intracellular nanobubbles.

Query 10:

How was the elimination pathway?

Answer 10:

We tried to evaluate the urine to evaluate the xenograft tumor-derived metabolites including NBO2. However, we could not show the clear difference between water group and NBO2 group. There is no direct evidence as to whether it is eliminated as oxygen molecules via exhalation or in the urine or feces as oxygen incorporated into metabolites.

This is also an important issue for future study.

Query 11:

Explain the criteria for choosing dose radiation of 3 Gy (HCT116) or 4 Gy (EBC-1) on days 4 and 8. It seems that was a part of a fractionated treatment.

Answer 11:

We performed a colony formation assay for checking cell survival and radiosensitization using modulation of the linear-quadratic parameters of the LQ model to be used in radiation dose calculations for in vivo experimentation. Based on the radiation dose-survival curves, we selected 3Gy per fraction for HCT116 and 4Gy for EBC-1 cells by two fractions in total.

Reviewer 2 Report

The authors aim to describe the effect of single-nanometer oxygen bubbles on radio sensitivity in tumor-bearing mice. The manuscript could be of potential interest. However, it presents several concerns and shows very few data to claim the effect of NBO2 as radiosensitizer. 

- the mechanism of radiosensitization proposed is based on the downregulation of HIF1alpha by NBO2, but in xenografts HIF1alpha expression after NBO+IR did not change as compared to NBO alone.

- Figure 4 does not indicate to which xenograft is referred HCT116 or EBC-1 or both?

-In supplementary figure 2 the WB is in duplicate while in figure 4 in triplicate and the housekeeping used were different.

-In HCB116 and ECB1 xenograft HIF1aplha expression was detected with different methods. It should be used the same assay or WB or IHC.

-The expression of HIF1alpha in ECB-1 cells is not showed

- To claim the radiosensitizing effect of NBO2, other hypoxia-related pathway must be investigated (i.e ROS, inflammation etc)

Author Response

Reply to Reviewer #2

Query 1:

The mechanism of radiosensitization proposed is based on the downregulation of HIF1alpha by NBO2, but in xenografts HIF1alpha expression after NBO+IR did not change as compared to NBO alone.

Answer 1:

Thank you for this suggestion. As the reviewer stated, the levels of HIF1 alpha in NBO2+IR group of Figure 4 and Supplementary figure 2 were not strongly suppressed compared to the NBO2 alone group. However, we believe that this result is consistent with our hypothesis because HIF1a levels in the NBO2+IR group were clearly suppressed compared to that in the Water group and Water+IR as control, and the suppression effect in the NBO2+IR group was induced by the combined NBO2.

Query 2:

Figure 4 does not indicate to which xenograft is referred HCT116 or EBC-1 or both?

Answer 2:

Thank you for this comment. We added the cell line information in Figure 4 and the Supplementary figure 2 legend.

Query 3:

In supplementary figure 2 the WB is in duplicate while in figure 4 in triplicate and the housekeeping used were different.

Answer 3:

We replaced the WB in supplementary figure 2 and revised the loading control according to this suggestion.

Query 4:

In HCB116 and ECB1 xenograft HIF1aplha expression was detected with different methods. It should be used the same assay or WB or IHC.

Answer 4:

Thank you for this suggestion. In supplementary Figure 2 of HCT116 and Figure 4 of EBC1, we showed the HIF-1α suppression effect by western blotting. We apologize for the lack of information in the Figure 4 legend. To clarify the cell line information, we have revised the figure legends. Moreover, we added the HIF-1α and pimonidazole staining in HCT116 xenograft tumors according to this suggestion (Supplementary figure 2).

Query 5:

The expression of HIF1alpha in ECB-1 cells is not showed

Answer 5:

We apologize for the confusing legend of Figure 4. It has been revised according to the previous query. We evaluated the HIF-1α expression in both EBC-1 and HCT116 cells (Figure 4 and Supplementary figure 2).

Query 6:

To claim the radiosensitizing effect of NBO2, other hypoxia-related pathway must be investigated (i.e ROS, inflammation etc)

Answer 6:

Thank you for this suggestion. We added the supplementary IHC data of carbonic anhydrase IX (CA IX) as a representative hypoxia related-downstream pathway in each treatment group. Many researchers already reported the importance of CAIXexpression on the radiation resistance in cancer cells. As expected, the expression levels of CA IX in the NBO2 water and NBO2 + IR groups were lower than those in the other groups. We added this new data as supplementary Figure 3.

Reviewer 3 Report

The authors present a paper about "Effects of ultrafine single mouse model".

The topic is interesting for its potential translational reasearch outcomes in the future.

At the moment the data presented is only in animal model.

Therefore the authors should:

1) explain their choice for the cell line

2) explain their choice for nude mouse xenograft model

3) explain the choice of 28 days for examination after administering NBO2 water

4) explain the choice for RT dose 

5) please modify the conclusion because the data presented is only about preclinical studies therefore before appluing such results to humans it will require additional incestigations.

6) it would interesting also to talk in the discussion section about the possible combination of radioprotective agents (such as amifostine and palifermin for example).

Author Response

Reply to Reviewer #3

Query 1:

The authors should explain their choice for the cell line

Answer 1:

We selected these cell lines because we have experience treating NBO2 or using the radiation treatment in the animal model. Previously, we used tumor-bearing mouse models to show the radiation sensitizer effect of the UBE2 inhibitor. Then we used HCT116 models (ANTICANCER RESEARCH 37: 2425-2436 (2017)). Regarding the EBC-1 cell line, we used the cell lines to evaluate the NBO2 effect in vitro analysis (Int J Oncol. 2018 Mar;52(3):679-686) and have experience in establishing the tumor-bearing xenograft model.

Query 2:

The authors should explain their choice for nude mouse xenograft model

Answer 2:

As mentioned above, we have the experience establishing the xenografted tumor mouse model to evaluate the radiosensitizer effect. Therefore, we selected the model in this study. In the next step, we want to clarify the relation between NBO2 treatment and immune cell activity in a syngeneic mouse model.

Query 3:

The authors should explain the choice of 28 days for examination after administering NBO2 water

Answer 3:

In this study, we accord to the previously established method to evaluate the radiosensitizer effect in a tumor bearing xenograft mouse model.

Query 4:  

The authors should explain the choice for RT dose.

Answer 4:

We performed a colony formation assay for checking cell survival and radiosensitization using modulation of the linear-quadratic parameters of the LQ model to be used in radiation dose calculations for in vivo experimentation. Based on the radiation dose-survival curves, we selected 3Gy per fraction for HCT116 and 4Gy for EBC-1 cells by two fractions in total.

Query 5:

Please modify the conclusion because the data presented is only about preclinical studies therefore before applying such results to humans it will require additional investigations.

Answer 5:

According to the suggestion, we deleted the overstatement regarding the potential of clinical application in the conclusion paragraph.

Query 6:

It would interesting also to talk in the discussion section about the possible combination of radioprotective agents (such as amifostine and palifermin for example).

Answer 6:

Using nanobubble oxygen water plus radiotherapy treatment combined with radioprotective agents such as amifostine may be interesting. However, in mice, harmful effects such as dry skin, body weight loss, decreased red blood cell count, or abnormal serum biochemical examinations were not detected with the nanobubble oxygen water until 28 days. Therefore, we did not check a combination of radiation protective drugs in this study. To explain the potential of the combined effect of nanobubble water with such agents, we added a new discussion section as follows:

"On the other hand, amifostine has been approved as a clinical radiation protector to reduce the radiation-related side effect in normal tissues [19]. Therefore, we expect that amifostine combined with NBO2 water could allow higher radiation doses, thereby enhancing the therapeutic effect without increasing radiation-related side effects.”

Round 2

Reviewer 2 Report

The authors partially answered to my comments, however the manuscript still presents serious concerns. Authors claim that their data demonstrate that NBO2 water suppressed hypoxia-induced radiation resistance in cancer cell lines via the downregulation of HIF-1α accumulation under hypoxic conditions, but both cell lines used seem not resistant to radiotherapy. How NBO2 suppresses radiation resistance in cells that are not resistant to this type of treatment? Probably would be more correct to state that NBO improve the efficacy of RT instead of suppressing hypoxia resistance.

The result section should be re-organized. Why the results relative to EBC-1 cells are included in the main text while the results relative to HCT116 are in the supplementary? All results should be reported within the main text. This manuscript report few results, it would be more correct to include all of them in the main manuscript avoiding supplementary figures.

The quality of WB (Suppl Fig 2) and of IHC (Fig 4) is poor. Histogram showing protein expression level in relation to houskeeping expression, should be included.

Author Response

Reply to Reviewer #2

Query 1:

The authors partially answered to my comments, however the manuscript still presents serious concerns. Authors claim that their data demonstrate that NBO2 water suppressed hypoxia-induced radiation resistance in cancer cell lines via the downregulation of HIF-1α accumulation under hypoxic conditions, but both cell lines used seem not resistant to radiotherapy. How NBO2 suppresses radiation resistance in cells that are not resistant to this type of treatment? Probably would be more correct to state that NBO improve the efficacy of RT instead of suppressing hypoxia resistance.

Answer 1:

As the reviewer indicated, we previously reported the HCT116 cell line as one of the radiation-sensitive cell lines. We agree with this comment. Therefore, according to this advice, we have revised the descriptions regarding the effect of our NBO2 water in the result and conclusion sections.

Query 2:

The result section should be re-organized. Why the results relative to EBC-1 cells are included in the main text while the results relative to HCT116 are in the supplementary? All results should be reported within the main text. This manuscript report few results, it would be more correct to include all of them in the main manuscript avoiding supplementary figures.

Answer 2:

Thank you for this suggestion. As suggested, we have described all results as major data and deleted the supplementary data.

Query 3:

The quality of WB (Suppl Fig 2) and of IHC (Fig 4) is poor. Histogram showing protein expression level in relation to housekeeping expression, should be included.

Answer 3:

As suggested, we have added a bar graph to show the protein expression levels.

Reviewer 3 Report

I have no further comments

Author Response

Reply to Reviewer #3

Query 1:

I have no further comments

Answer 1:

Thank you for this comment.